# Solar-Blind Ultraviolet Four-Quadrant Detector and Spot Positioning System Based on AlGaN Diodes

**DOI:** 10.3390/s25072206

**Published:** 2025-03-31

**Authors:** Longfei Peng, Shangqing Li, Yong Huang, Yang Yang

**Affiliations:** 1State Key Laboratory of Modern Optical Instrumentation, College of Optical Science and Engineering, Zhejiang University, Hangzhou 310007, China; 22430092@zju.edu.cn; 2Xidian-Wuhu Research Institute, Bldg. 7, Science and Technology Industrial Park, No. 717, Zhongshan (S) Rd., Yijiang District, Wuhu 241000, China; huangyong@xdwh-inst.com

**Keywords:** four-quadrant detector, solar-blind ultraviolet, AlGaN, spot positioning, photoelectric signal processing

## Abstract

The four-quadrant detector (4QD), as a highly sensitive and fast-response position-sensitive device, is widely used in laser guidance, target tracking, and related fields. However, traditional visible and infrared 4QDs exhibit significant vulnerability to ambient light interference, particularly under high-intensity background illumination. To address this issue, this paper presents a solar-blind ultraviolet (UV) 4QD and a spot positioning system based on AlGaN diodes, achieving a UV/visible suppression ratio of 2.17 × 10^4^ (without solar-blind filters). The system employs a high-linearity, low-noise capacitive transimpedance amplifier (CTIA) as the readout circuit for the high-sensitivity and rapid-response solar-blind UV detectors, enabling the precise conversion of weak photocurrent signals into voltage signals for digitization. Utilizing a third-order polynomial least-squares fitting algorithm without introducing complex filtering techniques, the system achieves a maximum positioning error of 0.0101 mm and a root-mean-square error (RMSE) of 0.0057 mm, among of one the best-performing solar-blind UV 4QDs. The experimental results demonstrate exceptional spot positioning performance under a 275 nm laser source, meeting the high-precision requirements for space target detection. This research provides a reference for the application of solar-blind UV 4QDs in positioning, alignment, and monitoring scenarios, thereby holding significant practical implications.

## 1. Introduction

Traditional visible and infrared detection methods are susceptible to interference from solar radiation and atmospheric background, making them unsuitable for all-day and all-weather target detection. In contrast, the solar-blind UV band (240~280 nm) is naturally “blind” to solar radiation due to strong atmospheric absorption, offering low background noise and superior anti-interference capabilities. These advantages make solar-blind UV detection highly promising for applications in space target tracking, missile warning systems, environmental monitoring, and scientific research [1,2,3].

The 4QD is a position-sensitive device with the advantages of high sensitivity, fast response speed, and high precision, which has a wide range of applications in laser guidance, target tracking, space optical communication, and other fields. Recent studies on 4QDs have mainly focused on improving detection accuracy, response speed, and anti-interference ability, and expanding its application fields. For example, Li et al. [4] proposed a cyclic cross-correlation method to enhance spot detection accuracy in free-space optical communication, achieving an RMSE of 0.0092 mm under an extremely low signal–noise ratio (SNR = −17.86 dB). Wang et al. [5] introduced error compensation coefficients into traditional models, reducing the maximum spot position error to 0.0277 mm (RMSE = 0.0065 mm) in high-background-noise environments. Zhang et al. [6] developed a segmented low-order polynomial least-squares fitting algorithm combined with a Kalman filter, which not only significantly improves detection accuracy but also achieves faster computation for real-time processing on microprocessors.

However, most reported 4QD systems operate in infrared or visible bands, which lack robustness under strong background light. Therefore, exploring the application of 4QDs in the solar-blind UV band is of significant importance.

This study presents a solar-blind UV spot positioning system based on a 4QD made of four AlGaN diodes. By combining the high sensitivity and ultrafast response of AlGaN solar-blind UV detectors with the low-noise, high-linearity characteristics of a CTIA, the system achieves rapid and accurate determination of target positions through the real-time analysis of the spot’s energy distribution across quadrants. The experimental results demonstrate that the system is expected to achieve high-precision tracking of dynamic targets, showcasing its potential for deployment in demanding applications such as real-time industrial process monitoring and alignment calibration of UV communication devices in optically cluttered environments.

## 2. Solar-Blind UV 4QD

As shown in Figure 1a, the 4QD comprises four independent square-shaped AlGaN-based UV detectors, each with a ~1 mm^2^ photosensitive area, resulting in an effective detection area of ~4 mm^2^. The overall device footprint of the 4QD is 7.44 mm. Adjacent photosensitive surfaces are separated by a non-sensitive gap termed the “dead zone”, whose width critically influences the total optical energy captured from the incident spot. Empirical studies [7] demonstrate that the detection sensitivity of the 4QD will increase with the increase in the ratio of the dead zone to the spot size.

The detector employs a quasi-vertical structure (QVS). As shown in Figure 1b, the QVS features a vertically stacked p-AlGaN/i-AlGaN/n-AlGaN design, complemented by metal electrodes fabricated on lateral facets to establish a unique quasi-vertical carrier transport pathway. In contrast to conventional vertical structures limited by UV photon absorption losses in transparent conductive electrodes (e.g., ITO/IZO), the QVS eliminates the need for such electrodes, achieving an extremely high UV photon penetration efficiency. This enhancement is facilitated by the direct exposure of the photosensitive AlGaN layer to incident radiation. Critically, the QVS design significantly enhances the external quantum efficiency (EQE), a key metric quantifying the detector’s ability to convert incident photons into photogenerated carriers. As illustrated in Figure 1c, the EQE reaches 56.2% at 275 nm under zero bias, a great improvement over traditional vertical structures (EQE < 20%).

The built-in electric field of the p-i-n junction enables rapid carrier separation, yielding a detector response time of 11.4 ns (Figure A1), thus satisfying the sub-microsecond detection requirements. Furthermore, under zero bias, the dark current density is suppressed to ~25 pA/mm^2^, which is beneficial for reduced noise current and thus more accurate position estimation, as discussed in Figure A2.

Prior to investigating the position detection characteristics of the 4QD, the spectral response of individual photodetector components was systematically characterized. As shown in Figure 1d, the spectral response of the photodetector under zero bias spans 200~370 nm, fully encompassing the solar-blind UV band (240~280 nm). A peak responsivity of 0.19 A/W is observed at 355 nm, attributed to the direct bandgap transition of the AlGaN active layer. The ultraviolet-to-visible (UV/vis) suppression ratio, defined as the responsivity ratio at 255 nm to 520 nm, serves as a critical metric for evaluating the detector’s spectral selectivity and resistance to background light interference. Additionally, the suppression ratio between 255 nm and 400 nm (255/400 nm) further quantifies rejection performance in the near-UV region. The experimental measurements yield a UV/vis suppression ratio of 2.17 × 10^4^ and a 255/400 nm suppression ratio of 2521, suggesting excellent visible blindness characteristics. To further enhance the detector’s UV/vis suppression ratio and near-UV rejection capability while preserving its high UV sensitivity, an interference filter with an optical density (OD) > 3 was integrated upstream of the detector. This solar-blind UV bandpass filter exhibits a sharp transmission window spanning 250~290 nm, coupled with deep cut-off characteristics (OD > 3) across the visible spectrum and near-UV region. As shown in Figure 1e,f, the filtered spectral response demonstrates exceptional selectivity, achieving a UV/vis suppression ratio of 5.79 × 10^5^ and a 255/400 nm suppression ratio of 4.34 × 10^4^, corresponding to 26.67-fold and 17.2-fold improvements over the unfiltered detector, while maintaining a high UV quantum efficiency of over 40% at 275 nm.

In addition to spectral selectivity, the dynamic range of the AlGaN photodetectors was evaluated to assess their capability to handle varying illumination intensities, a critical requirement for real-world applications such as high-speed target tracking. The photodetectors exhibit a linear responsivity spanning four orders of magnitude in optical power density (as shown in Figure A3) under 275 nm illumination. This wide dynamic range ensures robust performance across scenarios ranging from weak UV signals (e.g., distant targets) to high-flux conditions, which demonstrates the detector’s ability to adapt to environments with rapid fluctuations in solar-blind UV intensity.

## 3. Photoelectric Signal Processing Circuit

The photoelectric signal processing circuit transduces the weak photocurrent signals from the 4QD into voltage outputs for beam position determination. In this work, a capacitive feedback transimpedance amplifier (CTIA) is implemented as the front-end readout circuit, which converts photogenerated charges into quantifiable voltage signals through capacitive integration. Compared to conventional resistive feedback architectures, the CTIA topology offers distinct advantages in weak-signal amplification and high-precision metrology. Specifically, capacitive feedback eliminates the inherent limitations of resistive configurations, including thermal Johnson–Nyquist noise and long-term drift associated with high-impedance resistors, thereby enhancing system signal–noise ratio (SNR) and measurement fidelity. The operational principle of the CTIA is illustrated in Figure 2a: Upon UV illumination, photogenerated carriers induce a transient current *i*, which is integrated over time by capacitor *C*. The resultant output voltage *u* follows the relationship: *u = ∫idt/C = q/C*, where *q* denotes the total integrated charge. By differentially analyzing the voltage outputs from each quadrant, the spatial centroid of the incident UV beam can be precisely resolved.

The DDC114 (Texas Instruments Inc., Dallas, TX, USA) represents a monolithic 20-bit quad-channel current-input ADC that integrates four CTIAs at its input stage. This architecture simultaneously achieves current-to-voltage conversion and high-resolution digitization on-chip, eliminating the need for discrete front-end signal conditioning. By integrating the ADC with the photodetector readout stage, this architecture inherently suppresses noise contamination and signal integrity degradation associated with analog signal transmission, thereby enabling direct digital processing of photocurrents at the picoampere (pA) level.

As illustrated in Figure 2b, the DDC114’s input stage employs an innovative configuration of eight discrete integration capacitors with capacitance values spanning 3 pF to 87.5 pF. This multi-capacitor architecture enables current measurement across a nine-order-of-magnitude dynamic range, from femtoampere (fA) to microampere (µA) regimes. Range switching is accomplished via complex programmable logic device (CPLD)-controlled pin configuration, which selectively activates specific capacitors to adapt to varying illumination intensities. The device exhibits exceptional metrological performance: a noise floor of 5.2 parts-per-million (ppm) of full-scale (FS) and integral non-linearity (INL) of ±0.5 ppm FS, establishing its suitability for ultralow-current photometric applications.

The signal processing latency of the DDC114 is primarily governed by its programmable integration time, which spans from 320 μs to 1 s depending on the selected capacitor configuration and illumination intensity. Shorter integration times prioritize high-speed response for dynamic tracking scenarios but sacrifice SNR, whereas longer integration times enhance SNR for weak photocurrent detection (e.g., sub-pA regimes) at the cost of reduced frame rates. This trade-off is dynamically managed through the CPLD-controlled capacitor switching logic, enabling adaptive optimization for real-time applications. This study implements a DDC114-based signal processing system (each module that makes up the system is shown in Figure 3a,b and the system is shown in Figure 3c) where weak photocurrents from the 4QD are routed to the DDC114 input stage via a flexible printed circuit (FPC) flat cable, enabling current-to-voltage integration and analog-to-digital (A/D) conversion under the timing control of CPLD. The system employs an Altera EPM570 CPLD (Intel Corporation, San Jose, CA, USA) as the central controller, integrated with an IS61LV25616AL (Integrated Silicon Solution Inc., San Jose, CA, USA) static random-access memory (SRAM) for data buffering and an MAX13488 (Analog Devices, San Jose, CA, USA) transceiver for host communication. By leveraging the CPLD’s reconfigurable logic and low-latency control, the architecture optimizes signal integrity and scalability for precision photometric applications.

## 4. Experimental Results

As illustrated in Figure 4, an experimental platform for spot position detection was established to investigate the positioning characteristics of the 4QD. A laser source with a wavelength of 275 nm was employed to illuminate the 4QD active area, adjusted in intensity by attenuators. Photogenerated currents from each quadrant were conditioned through CTIAs (with an integration time of 20 ms), producing quantifiable voltage outputs proportional to incident photon flux. These analog signals underwent 20-bit A/D conversion at 3.125 kS/s sampling rates prior to serial port transmission (1.152 Mbps baud rate) with a transmission latency of ~56 μs for computational processing.

To quantify the positional detection accuracy of the 4QD system, the device was mounted on a dual-axis micro-displacement platform with an accuracy of 0.5 μm. Since the *X*- and *Y*-axes are independent of each other in the calculation of the spot position, we analyzed the detection accuracy of the laser spot on the *X*-axis separately. A laser spot of a specific size was illuminated onto the 4QD (as shown in Figure A4). As the 4QD moved along the *X*-axis, the spot position was calculated using Formulas (A1) and (A2) in Appendix B. The linear relationship between the calculated spot positions and the actual displacement is demonstrated in Figure 5a. Within the vicinity of the center of the 4QD, the curve exhibits near-ideal linearity, with calculated spot coordinates deviating very little from theoretical predictions based on geometric centroid algorithms. Beyond this linear operating range, non-linear distortion gradually emerges due to the truncation of the Gaussian beam at the periphery of the detector, resulting in a non-linear relationship between position error and displacement. Systematic characterization reveals a peak absolute positional error of 0.2519 mm, accompanied by a root-mean-square error (RMSE) of 0.1 mm across the full measurement span. In order to demonstrate the non-linearity of the positional response more intuitively, the spot position detection error is also given in Figure 5b.

While conventional spot position calculation models derived from Equations (A1) and (A2) demonstrate high computational efficiency, their inherent accuracy limitations render them inadequate for precision-critical applications. Recent advancements in position detection algorithms-including piecewise low-order polynomial least squares fitting [6], integral infinite log-ratio optimization [8], and convolutional neural network architectures [9,10,11,12]-have addressed these constraints at the cost of increased computational overhead. Through systematic analysis of the non-linear error distribution in Figure 5a, this work implements an improved algorithm based on piecewise low-order polynomial least squares fitting. The curve is partitioned into four segments, with each segment undergoing localized cubic polynomial fitting via least-squares minimization:(1)$$x=\left\{\begin{array}{c} 21.867{\delta_{x}}^{3}-48.871{\delta_{x}}^{2}+38.11\delta_{x}-8.347\;\;(\delta_{x}>0.53)\\ -1.9734{\delta_{x}}^{3}+1.4965{\delta_{x}}^{2}+2.3504\delta_{x}+0.0008\;\;\;\left(0\leq \delta_{x}\leq 0.53\right)\\ 0.8486{\delta_{x}}^{3}-2.411{\delta_{x}}^{2}+2.1096\delta_{x}-0.0047\;\;\left(-0.53\leq \delta_{x}<0\right)\\ 6.0384{\delta_{x}}^{3}+12.621{\delta_{x}}^{2}+10.253\delta_{x}+1.1552\;\;(\delta_{x}<-0.53) \end{array}.\right.$$where *δ_x_* represents the normalized coordinate of the *X*-axis, which is defined in Equation (A1).

Comparative analysis between linear models (Figure 5a,b) and the optimized piecewise polynomial algorithm (Figure 5c,d) demonstrates substantial error mitigation: the maximum absolute positional error exhibits a 95.9% reduction (from 0.2519 mm to 0.0101 mm), while the root-mean-square error (RMSE) decreases by 94.3% (original: 0.1 mm; optimized: 0.0057 mm).

For a better comparison, Table 1 lists the core parameters of quadrant detectors based on different solar-blind UV semiconductor materials, mainly concerning responsivity, UV/visible rejection ratio, and positioning error. The experimental data show that the AlGaN 4QDs demonstrated in this paper have very good visible blindness performance and high positioning accuracy, which is of great significance in bringing the solar-blind UV 4QD technology to practical applications.

## 5. Conclusions

This study developed a solar-blind UV 4QD-based spot positioning system made of AlGaN diodes, with a UV/visible rejection ratio of 2.17 × 10^4^ (without solar-blind filters). Using a third-order polynomial least-squares fitting algorithm, the system achieved a maximum error of 0.0101 mm and RMSE of 0.0057 mm without complex noise suppression techniques. This research provides a conceptual design for utilizing solar-blind UV 4QD in engineering applications, including positioning, alignment, and monitoring.

There are also limitations to the research in this paper. The current study validates the system’s optoelectronic performance and position detection characteristics under laboratory conditions but does not assess its resilience to harsh environments. For example, extreme temperature fluctuations (−50 °C to +150 °C) in space applications or high-energy particle radiation in near-Earth orbits could reduce the performance of the 4QD. These limitations provide a clear roadmap for advancing the technology toward real-world deployment.

## Figures and Tables

**Figure 1 sensors-25-02206-f001:**
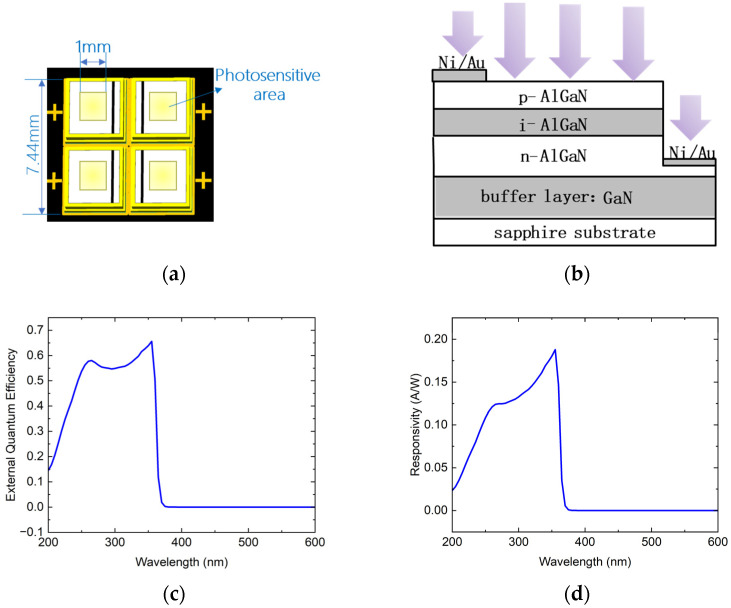
(**a**) Schematic diagram of the four-quadrant detector; (**b**) schematic of the quasi-vertical structure UV photodetector; (**c**) external quantum efficiency curve of the UV photodetector without a filter; (**d**) photoresponsivity curve of the UV photodetector without a filter; (**e**) external quantum efficiency curve of the UV photodetector integrated with a 250~290 nm bandpass solar-blind UV filter; (**f**) photoresponsivity curve of the UV photodetector integrated with a 250~290 nm bandpass solar-blind UV filter.

**Figure 2 sensors-25-02206-f002:**
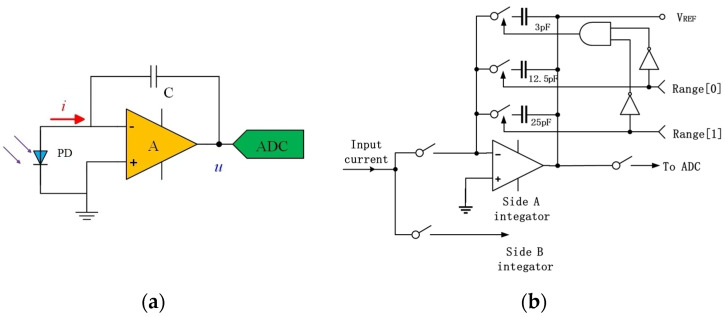
(**a**) Schematic of the capacitive feedback transimpedance amplifier (CTIA); (**b**) schematic diagram of the input-stage structure of the DDC114.

**Figure 3 sensors-25-02206-f003:**
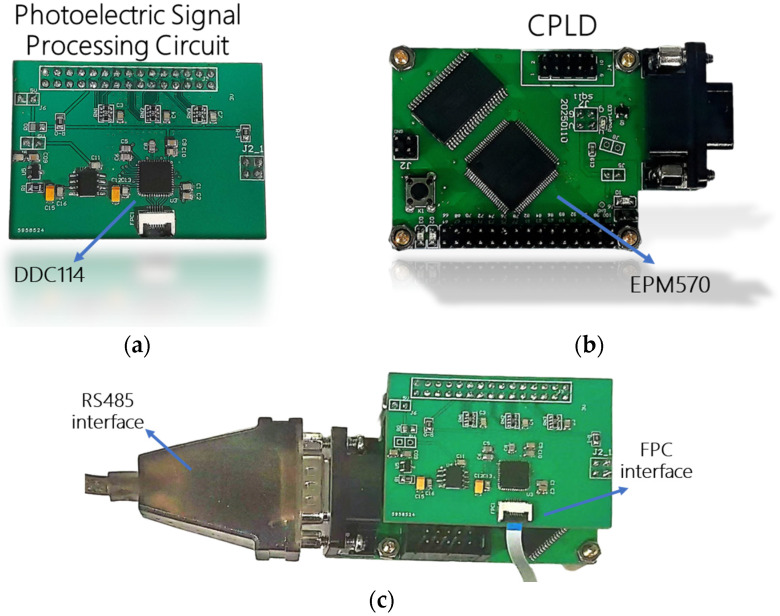
(**a**) Signal acquisition module based on DDC114; (**b**) master control module based on EPM570; (**c**) overall structure of signal processing system.

**Figure 4 sensors-25-02206-f004:**
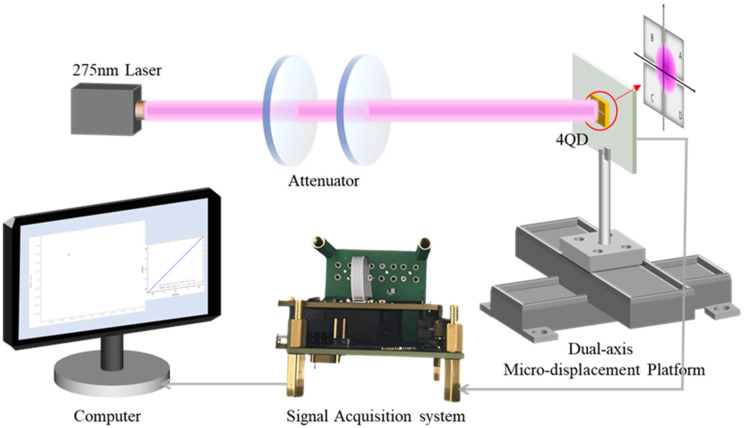
Schematic diagram of a solar-blind UV four-quadrant detector (4QD) based spot positioning system. The system consists of a laser beam emitted by a 275 nm laser, adjusted in intensity by attenuators, and directed to the 4QD, which is mounted on a dual-axis micro-displacement platform; the Signal Acquisition System collects the detector signals and transmits them to a computer for analysis and processing to achieve spot positioning.

**Figure 5 sensors-25-02206-f005:**
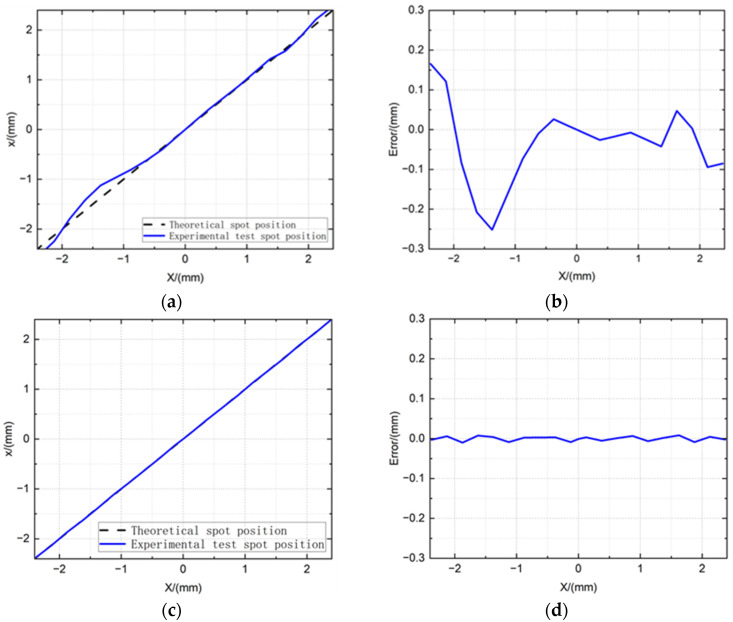
(**a**) Calculation results of the *X*-axis position of the laser spot based on the conventional algorithm; (**b**) calculation error of the *X*-axis position of the laser spot based on the conventional algorithm; (**c**) calculation results of the *X*-axis position of the laser spot based on the third-order polynomial fitting; (**d**) calculation error of the *X*-axis position of the laser spot based on the third-order polynomial fitting.

**Table 1 sensors-25-02206-t001:** Summary of some key parameters of solar-blind quadrant detectors.

Material	Responsivity (A/W)	UV/Visible Rejection Ratio	Detection Range (mm)	RMSE (mm)	Mean Absolute Error (mm)	Ref.
Ga_2_O_3_ PSD	~6 × 10^−5^ (@243 nm)	133 (243/400 nm)	—	—	—	[13]
AlGaN 4QD	~0.028 (@250 nm)	10^3^~10^5^ (250/≥400 nm)	—	—	—	[14]
4H-SiC 4QD	~0.111 (@275 nm)	>10^3^ (275/400 nm)	[−0.8, 0.8]	—	0.0285	[15]
4H-SiC 4QD	~0.12 (@266 nm)	>2000 (275/400 nm)	[−0.3, 0.3]	—	0.0055	[16]
AlGaN 4QD	0.13 (@275 nm)	2.17 × 10^4^ (255/520 nm, NF) 2521 (255/400 nm, NF) 5.79 × 10^5^ (255/520 nm, F) 4.34 × 10^4^ (255/400 nm, F)	[−0.5, 0.5]	—	0.0036	This work
			[−2.4, 2.4]	0.0057	0.0050	

NF for no filter added; F for solar-blind UV filter added.

## Data Availability

The data related to this study can be requested from the corresponding authors.

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
