# Peer review of "Solar-Blind Ultraviolet Four-Quadrant Detector and Spot Positioning System Based on AlGaN Diodes"

_sensors, 2025, doi:10.3390/s25072206_

Round 1
Reviewer 1 Report
Comments and Suggestions for Authors
This manuscript presents an innovative solar-blind UV spot positioning system based on a 4QD made of AlGaN diodes. The system demonstrates high positioning accuracy through experimental validation and optimization of a third-order polynomial fitting algorithm.By integrating high-sensitivity AlGaN detectors with a capacitive transimpedance amplifier (CTIA) readout circuit, the system demonstrates exceptional positioning accuracy under 275 nm UV laser illumination. overall I recommentd this manuscript to be published, and I have a few technical questions.
1. The dynamic range of the AlGaN diodes can be added and discussed in the manuscript.
2. the author can add more discussions on the system's limitations and potential areas for future improvement.
Reviewer 2 Report
Comments and Suggestions for Authors
This manuscript introduces a solar-blind UV four-quadrant detector based on AlGaN diodes. 4QD based on AlGaN diodes has been investigated long time ago which could address the background noise limitations. Although AlGaN-based 4QD systems have indeed been under investigation for a while, this manuscript appears to offer new contributions in its system-level integration and performance.
Comments:
- It is not necessary to have a separate section of 4QD fundamental working principle in the main manuscript. I suggest moving it to supporting information. Since this manuscript is focusing on system integration, I would suggest to have 4QD as one separate section (section 2), then a section of description of signal acquisition system (section 3), system performance (section 4).
- Page 4, line 103, how is this AlGaN photodiode prepared? Author did not mention the preparation method
- Figure 2. Subfigure a&b: need to enlarge font size to make it clear. Currently it is hard to read.
- Figure 2. Subfigure c&f: the border of figure is visible. Need to be removed.
- Figure 3. Subfigure a&b: the border of figure is visible. Need to be removed. Figure resolution also low. Need to re-adjust.
Reviewer 3 Report
Comments and Suggestions for Authors
The manuscript presents a high-performance solar-blind ultraviolet (UV) four-quadrant detector (4QD) based on AlGaN diodes, integrated into a spot positioning system. The presentation and delivery of this work is clear, and the manuscript is well structured. The reviewer would highly recommend the acceptance of this work after minor revisions.
i) Since the Introduction part mentioned such solar-blind photodetectors may be potentially used in harsh conditions, could the authors comment on the robustness of your detection system when implementing in these demanding environments?
ii) In the benchmark table, the reviewer is curious whether any commercial system is available for comparison?
iii) Could the authors consider discussing the latency introduced by signal processing, transmission, and computation, which is critical for real-time applications?
iv) In Fig.2, it is suggested that Fig. 2(e) and (f) be switched for a better visual correlation with the order of (c) and (d).
v) The fitting model introduces segment-wise corrections, indicating that the detector's response may not be perfectly uniform across all quadrants. Could the authors comment on this?
vi) Lastly, how were the photodetectors fabricated? Were they produced in a university cleanroom? Or were they commercially sourced?
